# Sex-specific resilience of neocortex to food restriction

**Zahid Padamsey[1,2]\*, Danai Katsanevaki[2,3], Patricia Maeso[2], Manuela Rizzi[2], Emily E Osterweil[2,3,4], Nathalie L Rochefort[2,3]\***

[1]Wellcome-MRC Institute of Metabolic Science, University of Cambridge, Cambridge, United Kingdom; [2]Centre for Discovery Brain Sciences, School of Biomedical Sciences, University of Edinburgh, Edinburgh, United Kingdom; [3]Simons Initiative for the Developing Brain, University of Edinburgh, Edinburgh, United Kingdom; [4]Rosamund Stone Zander Translational Neuroscience Center, F.M. Kirby Center, Boston Children's Hospital, Harvard Medical School, Boston, United States

**\*For correspondence:**
zp278@cam.ac.uk (ZP);
n.rochefort@ed.ac.uk (NLR)

**Competing interest:** The authors declare that no competing interests exist.

**Abstract** Mammals have evolved sex-specific adaptations to reduce energy usage in times of food scarcity. These adaptations are well described for peripheral tissue, though much less is known about how the energy-expensive brain adapts to food restriction, and how such adaptations differ across the sexes. Here, we examined how food restriction impacts energy usage and function in the primary visual cortex (V1) of adult male and female mice. Molecular analysis and RNA sequencing in V1 revealed that in males, but not in females, food restriction significantly modulated canonical, energy-regulating pathways, including pathways associated waith AMP-activated protein kinase, peroxisome proliferator-activated receptor alpha, mammalian target of rapamycin, and oxidative phosphorylation. Moreover, we found that in contrast to males, food restriction in females did not significantly affect V1 ATP usage or visual coding precision (assessed by orientation selectivity). Decreased serum leptin is known to be necessary for triggering energy-saving changes in V1 during food restriction. Consistent with this, we found significantly decreased serum leptin in food-restricted males but no significant change in food-restricted females. Collectively, our findings demonstrate that cortical function and energy usage in female mice are more resilient to food restriction than in males. The neocortex, therefore, contributes to sex-specific, energy-saving adaptations in response to food restriction.

## eLife assessment

This study provides **important** findings based on **compelling** evidence demonstrating that females and males have different strategies to regulate energy consumption in the brain in the context of low energy intake. While food deprivation reduces energy consumption and visual processing performance in the visual cortex of males, the female cortex is unaffected, likely at the expense of other functions. This study is relevant for scientists interested in body metabolism and neuroscience.

## Introduction

Mammals reduce their energy usage in times of food scarcity. These adaptations have been well documented for peripheral tissues and are sex specific. For example, females, as compared to males, readily lose muscle and bone mass to reduce peripheral energy expenditure during food restriction, but lose less fat mass and overall bodyweight (*Andersson et al., 1991*; *Cortright and Koves, 2000*; *Volek et al., 2004*; *McCarthy and Berg, 2021*; *Tirosh et al., 2015*; *Suchacki et al., 2023*). Moreover, females are more likely to suppress energy-costly, reproductive functions during food restriction than

males, evidenced by substantive reductions in uterine and ovarian mass, as well as a cessation of reproductive function and behaviour (*Boutwell et al., 1948*; *Ahima et al., 1996*; *Gill and Rissman, 1997*). Such sex-specific differences are thought to reflect the differential importance of different organs to sex-specific survival (*Cortright and Koves, 2000*).

In contrast to peripheral tissue, whether and how the brain adapts its function and energy usage in a sex-specific manner during food scarcity remains largely unknown. The brain requires substantial amounts of energy to process and encode information (*Harris et al., 2012*; *Attwell and Laughlin, 2001*; *Herculano-Houzel, 2011*). Indeed, the human brain, which comprises 2% of our body's mass, consumes 20% of our caloric intake, over half of which is used by the cerebral cortex (*Herculano-Houzel, 2011*). Previously, we demonstrated that food restriction in male mice reduced energy usage in the primary visual cortex (V1) during visual processing, which was associated with a loss of orientation selectivity and visual coding precision (*Padamsey et al., 2022*). These changes required a decrease in levels of serum leptin, a hormone secreted by adipocytes in proportion to fat mass (*Baile et al., 2000*). Thus, in males the cortex, like peripheral tissue, has the capacity to reduce its energy usage and function in times of food scarcity.

In contrast, it remains unclear to what extent the brain similarly contributes to energy-saving adaptations during food scarcity in mammalian females. Understanding how food restriction impacts cortical function in males and females is not only of fundamental importance for understanding the sex-specific impact of diet on brain function, but is also critical for studies of cortical function, in which food restriction is used to behaviourally motivate animals (*Guo et al., 2014*; *Goltstein et al., 2018*; *Toth and Gardiner, 2000*).

Here, we examined how food restriction leading to a 15% loss in bodyweight over 2–3 weeks impacts energy usage and visual coding in the primary visual cortex (V1) in adult male and female mice. We found that leptin levels, which regulate energy-saving changes in *Padamsey et al., 2022*, were maintained in females during food restriction, despite being decreased by approximately three-fold in males. Consistent with these results, and using a range of experimental techniques, we failed to find significant energy-saving changes in V1 during food restriction in female mice, in contrast to male mice. Specifically, molecular analysis and RNA sequencing of V1 revealed that food restriction significantly modified canonical energy-regulating cellular pathways in males, but not in females, including pathways associated with AMP-activated protein kinase (AMPK), peroxisome proliferator-activated receptor alpha (PPARα), and mammalian target of rapamycin (mTOR) signalling, as well as oxidative phosphorylation. Moreover, using in vivo ATP and calcium imaging, we found that food restriction significantly reduced V1 ATP usage and orientation selectivity in males, but not in females.

Collectively, our study demonstrates that in times of food restriction, visual cortical function and energy usage are largely maintained in female mice, while they are reduced in males. The neocortex, therefore, contributes to sex-specific, energy-saving adaptations in response to metabolic stress. Our research highlights the importance of taking into account sex differences when studying the impact of dietary manipulations on brain function.

## Results

We first examined sex-specific differences in how the neocortex adapts to food restriction in mice. Mice, 7–9 weeks of age, either had ad libitum access to food (control group, CTR) or were food restricted (FR) to 85% of their free feeding bodyweight, for 2–3 weeks (*Figure 1A*). Prior to any of our recordings, tissue collection, and analysis, all animals were given ad libitum access to food until sated. This enabled us to examine the long-term impact of caloric restriction on neocortical function, as opposed to the short-term impacts of hunger (*Burgess et al., 2018*; *Burgess et al., 2016*).

### Sex-specific impact of food restriction on weight loss and fat mass-regulated hormone leptin

Prior to food restriction, we found that both males and females consumed the same ad libitum daily food intake, despite females weighing approximately 80% of age-matched males (weight: male vs. female; 27.82 g [95% confidence interval, CI: 26.84–28.79 g] vs. 23.21 g [95% CI: 22.35–24.07 g]; $t$ = 7.58; df = 66; p < 0.001; $n$ = 8 male and 7 female mice). Critically, we found that to achieve a similar magnitude (15%) and rate of weight loss in both sexes we had to impose a 40% greater restriction of

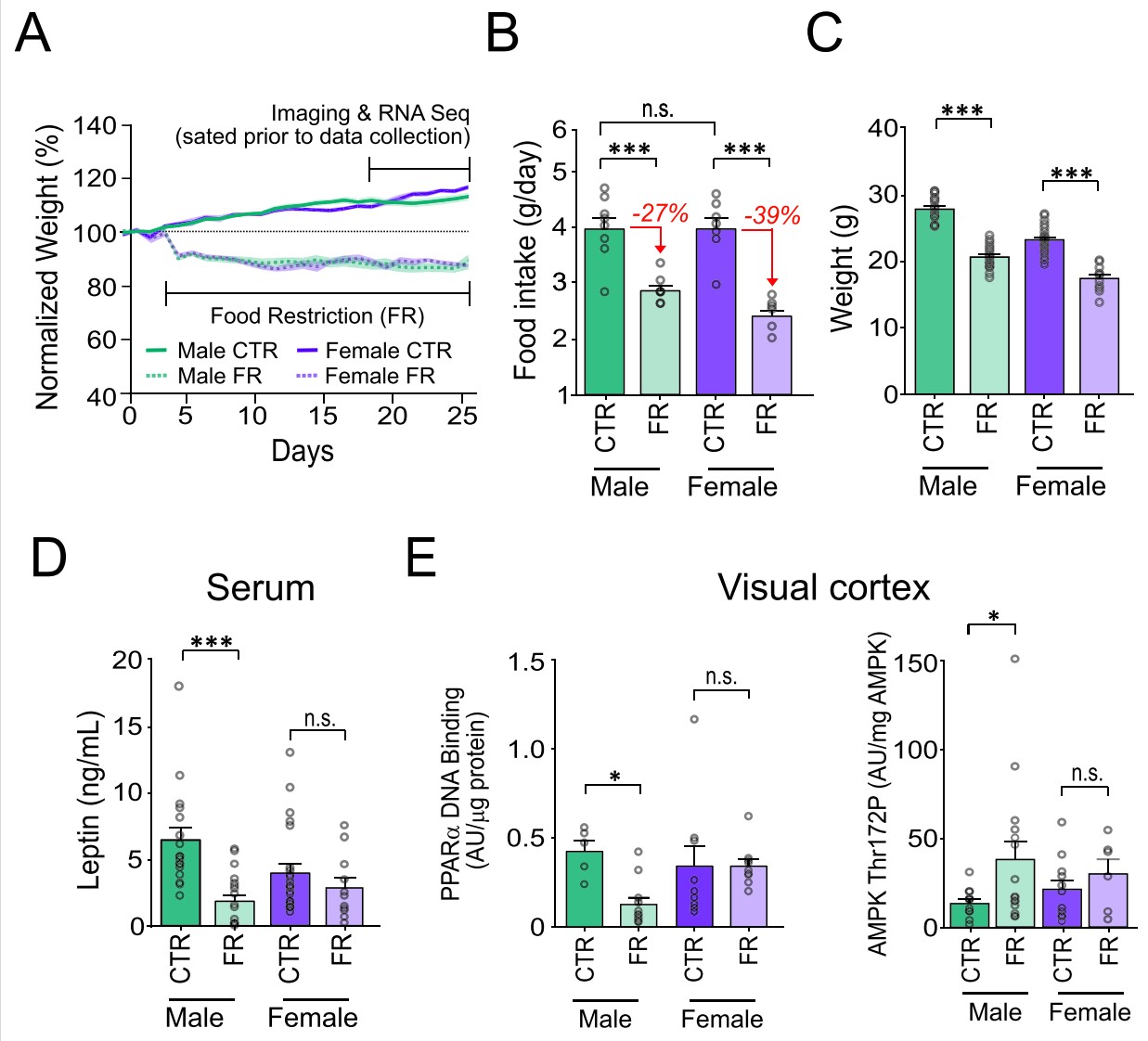

**Figure 1.** Sex-specific impact of food restriction on weight loss and serum leptin. (**A**) Animal weight across time. (**B**) Daily food intake (two-way analysis of variance [ANOVA]: CTR male vs. FR male; $t$ = 4.81; df = 25; p < 0.0001; CTR male vs. CTR female; $t$ = 0.013; df = 25; p = 0.99; CTR female vs. FR female; $t$ = 6.59; df = 25; p < 0.0001; FR male vs. FR female; $t$ = 1.93; df = 25; p = 0.07; $n$ = 8 CTR males, 8 FR males, 7 CTR females, and 7 FR females). Percent reduction of food intake for food restriction is shown in red for each sex. (**C**) Animal weight (two-way ANOVA: CTR male vs. FR male; $t$ = 11.36; df = 25; p < 0.0001; CTR female vs. FR female; $t$ = 8.36; df = 25; p < 0.0001; $n$ = 17 CTR males and 19 FR males; 23 CTR females and 11 FR females). (**D**) Serum leptin levels (two-way ANOVA: CTR male vs. FR male; $t$ = 4.58; df = 66; p < 0.0001; CTR females vs. FR females; $t$ = 1.00; df = 66; p = 0.32; $n$ = 17 CTR males and 19 FR males; 23 CTR females and 11 FR females). (**E**) Left: AMP-activated protein kinase (AMPK) Thr172 phosphorylation, normalized by total AMPK, in primary visual cortex (V1) tissue (two-way ANOVA: CTR male vs. FR male; $t$ = 2.28; df = 39; p = 0.022; CTR female vs. FR female; $t$ = 0.64; df = 39; p = 0.11; $n$ = 11 CTR males, 15 FR males, 11 CTR females, and 6 FR females). Right: Peroxisome proliferator-activated receptor alpha (PPARα) activity in V1 tissue, as assessed by levels of DNA binding, normalized to protein level (two-way ANOVA: CTR male vs. FR male; $t$ = 4.81; df = 30; p = 0.013; CTR female vs. FR female; $t$ = 0.0016; df = 30; p = 0.99; $n$ = 5 CTR males, 11 FR males, $n$ = 9 CTR females, and 9 FR females). ***p < 0.0001; *p < 0.05; n.s. = not significant. Error bars are standard error of the mean (SEM).

daily food intake in females (average of 39% of ad libitum intake; $n$ = 7 mice) compared to males (27% of ad libitum intake; $n$ = 8 mice) (**Figure 1B, C**). This is consistent with previous studies demonstrating that female mice are more resistant to weight loss than male (**Andersson et al., 1991**; **Cortright and Koves, 2000**; **Volek et al., 2004**; **McCarthy and Berg, 2021**; **Tirosh et al., 2015**; **Suchacki et al., 2023**). Moreover, for similar bodyweight loss (15%) maintained over 2–3 weeks, serum leptin levels, which reflect fat mass, were markedly and significantly decreased (−72%) in males (leptin: CTR male vs. FR male; 6.45 ng/ml [95% CI: 4.47–8.43 ng/ml] vs. 1.84 ng/ml [95% CI: 0.90–2.79 ng/ml]; $t$ = 4.58;

df = 66; p < 0.0001; $n$ = 17 CTR males and 19 FR males), but only modestly and non-significantly decreased (−28%) in females (leptin: CTR females vs. FR females; 3.94 ng/ml [95% CI: 2.53–5.35 ng/ml] vs. 2.83 ng/ml [95% CI: 1.20–4.47 ng/ml]; $t$ = 1.00; df = 66; p = 0.32; 23 CTR females and 11 FR females) (*Figure 1D*). This is in keeping with previous findings that female mice are more resistant to fat loss than males (*Andersson et al., 1991*; *Cortright and Koves, 2000*; *Volek et al., 2004*; *McCarthy and Berg, 2021*; *Tirosh et al., 2015*; *Suchacki et al., 2023*) and that weight loss arises from other tissues including those associated with reproductive functions (ovarian and uterine mass) (*Boutwell et al., 1948*; *Ahima et al., 1996*; *Gill and Rissman, 1997*).

## Cellular energy-regulating pathways in cortex are significantly impacted by food restriction in males but not in females

Decreased leptin levels are necessary for triggering energy-saving changes in the mouse visual cortex during food restriction (*Padamsey et al., 2022*). We therefore asked whether the absence of significant change in serum leptin in food-restricted females was associated with any change in V1 energy usage. We first assessed this at the molecular level by examining AMPK phosphorylation state. AMPK is a canonical cellular energy sensor activated by metabolic stress – such as food restriction, in part via decreased leptin signalling – through phosphorylation of threonine 172 (Thr$^{172}$) (*Garza-Lombó et al., 2018*; *Dagon et al., 2012*; *Dagon et al., 2005*). Its activation results in reduced ATP expenditure, for example, through the inhibition of mTOR signalling (*Garza-Lombó et al., 2018*; *Takei and Nawa, 2014*), and regulation of mitochondrial oxidative phosphorylation (*Herzig and Shaw, 2018*; *de la Cruz López et al., 2019*). In V1, we found that AMPK Thr$^{172}$ phosphorylation was markedly and significantly elevated (2.9-fold) with food restriction in males (AMPK Thr$^{172}$ phosphorylation: CTR male vs. FR male; 13.19 AU/µg [95% CI: 7.68–18.71 AU/µg] vs. 38.21 AU/µg [95% CI: 16.50–59.93 AU/µg]; $t$ = 2.28; df = 39; p = 0.022; $n$ = 11 CTR males and 15 FR males), consistent with ATP savings, but only modestly increased (1.4-fold) in females; this increase was not statistically significant (AMPK Thr$^{172}$ phosphorylation: CTR female vs. FR female; 20.94 AU/µg [95% CI: 9.42–32.46 AU/µg] vs. 29.52 AU/µg [95% CI: 8.52–50.53 AU/µg]; $t$ = 0.64; df = 39; p = 0.11; 11 CTR females, and 6 FR females) (*Figure 1E*). We also examined the activity of PPARα, which is downstream of leptin signalling and is known to regulate fatty acid metabolism and energy homeostasis (*Dagon et al., 2005*; *Unger et al., 1999*). As with AMPK, we found that PPARα was significantly regulated by food restriction selectively in males, but not in females (*Figure 1E*; *Poulsen et al., 2012*; *Wójtowicz et al., 2020*).

To further investigate how energy-regulating cellular pathways are impacted by food restriction, we performed RNA sequencing on tissue dissected from V1 (*Figure 2A*; $n$ = 4 CTR male, $n$ = 4 FR male, $n$ = 4 CTR female, and $n$ = 4 FR female mice). Differential gene expression analysis using DESeq2 revealed that food restriction significantly altered the expression of 657 and 585 targets in males and females, respectively, at a threshold of p-adj ≤0.1; only 125 of these were common to both sexes (*Figure 2B*). To assess the functional relevance of the transcriptomic changes induced by food restriction, we performed Gene Set Enrichment Analysis (GSEA) to identify pathways enriched in the differentially expressed populations. We found that food restriction resulted in a significant alteration of 34 and 14 gene sets (at p-adj ≤0.05) in males and females, respectively; 13 of these were common to both sexes (*Figure 2C*; *Supplementary file 1*; *Supplementary file 2*). In male mice, we specifically found that food restriction significantly regulated gene sets associated with canonical energy-regulating pathways (*Figure 2*), including those associated with oxidative phosphorylation and mTOR signalling (mTORC1 signalling and Phosphatidylinositol 3-kinase (PI3K)/Akt (protein kinase B)/mTOR signalling), as well as fatty acid metabolism; these pathways are known to regulated by leptin and AMPK signalling (*Garza-Lombó et al., 2018*; *Poulsen et al., 2012*; *Wójtowicz et al., 2020*; *Kwon et al., 2016*; *Figure 2D*). Notably, these pathways were not significantly regulated by food restriction in females. In addition to these findings, we found that gene sets associated to oestrogen-regulating pathways were significantly regulated in food-restricted males and not in females (*Figure 2C*). This is consistent with previous findings showing that oestrogen, like leptin, suppresses food intake and promotes energy expenditure (*Brown and Clegg, 2010*). Finally, consistent with past studies in cortex, we found that food restriction in males altered gene expression associated with inflammation and reactive oxidative species (*Figure 2D*; *Estep et al., 2009*; *Wood et al., 2015*).

Collectively, these findings reveal that food restriction strongly affects energy-regulating cellular pathways in the cortex of male, as opposed to female, mice.

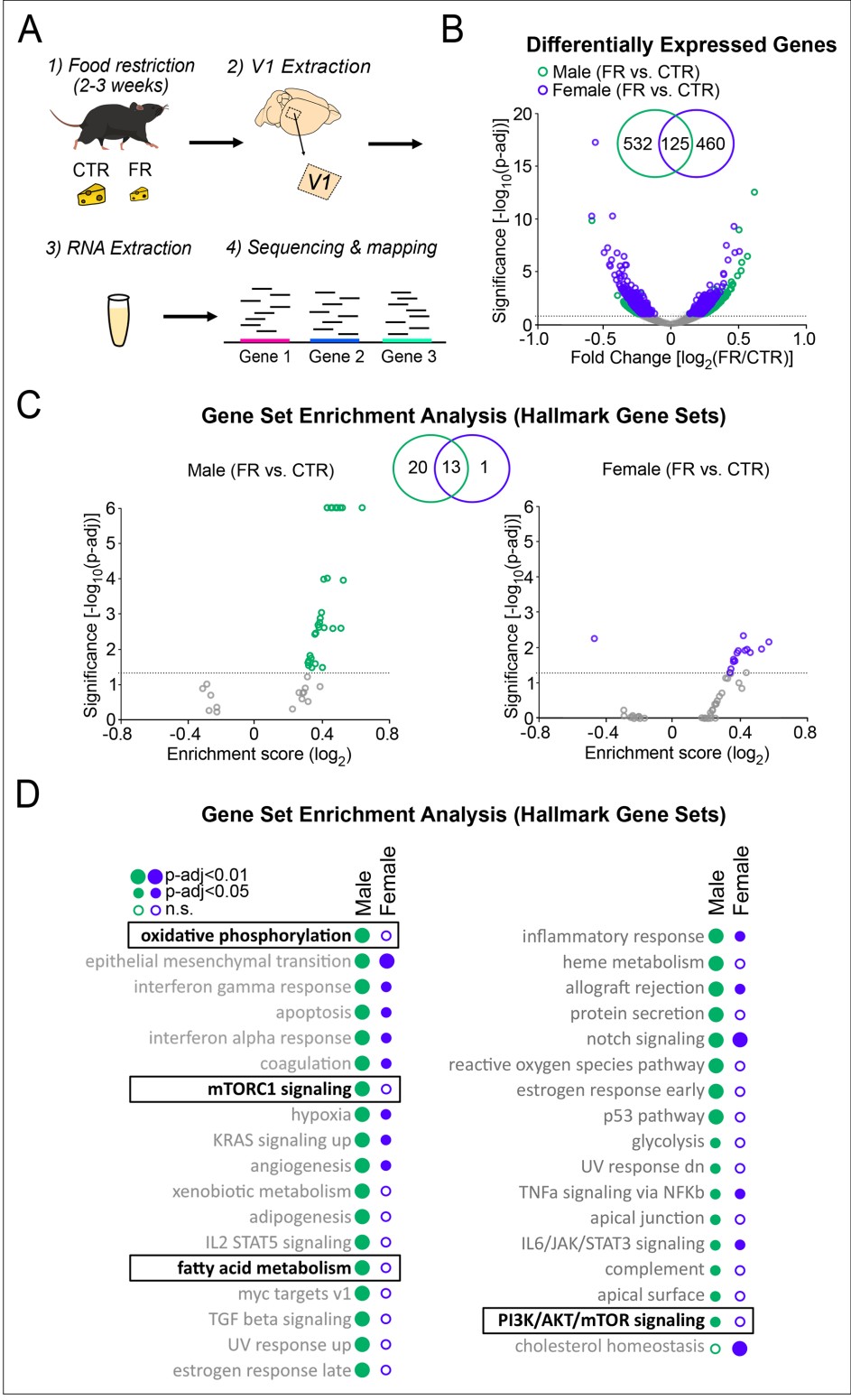

**Figure 2.** Cellular energy-regulating pathways are more robustly impacted by food restriction in males than in females. (**A**) Schema of RNA sequencing of V1 tissue. (**B**) Volcano plot of fold change expression with food restriction vs. significance for all analysed genes; broken horizontal line denotes adjusted p < 0.1 significance level, below which non-significantly regulated expressed genes are marked by grey. Inset: Venn diagram showing the number of significantly differentially expressed genes with food restriction in males and females. (**C**) Gene Set Enrichment Analysis using Hallmark Gene Sets, depicted using a volcano plot of Enrichment Score vs. significance

*Figure 2 continued on next page*

*Figure 2 continued*

for males (left) and females (right); broken horizontal lines denotes adjusted p < 0.05 significance level. Inset: Venn diagram showing the number of significantly regulated gene sets with food restriction in males and females. (**D**) Complete list of significantly enriched gene sets by food restriction in males and females for analysis in C. Gene sets of interest are denoted by a box (oxidative phosphorylation: male; *t* = 0.48; p < 0.0001; female; *t* = −0.17; p = 0.96; mTORC1 signalling: male; *t* = 0.46; p < 0.0001; female; *t* = −0.18; p = 0.98; fatty acid metabolism: male; *t* = 0.43; p < 0.0001; *t* = 0.2; p = 0.88; PI3K/AKT/mammalian target of rapamycin (mTOR) signalling: male; *t* = 0.34; p = 0.027; female; *t* = 0.17; p = 0.98). All data from four CTR males, four FR males, four CTR females, and four FR females.

## ATP usage in V1 is significantly decreased under food restriction in males but not in females

We next examined energy usage in visual cortex during visual processing using two-photon ATP imaging in V1 of awake, head-fixed Thy1-ATeam1.03$^{YEMK}$ mice, which express a Fluorescence Resonance Energy Transfer (FRET)-based ATP sensor (*Trevisiol et al., 2017*; *Baeza-Lehnert et al., 2019*; *Figure 3A*). In the presence of ATP synthesis inhibitors, focally applied over the visual cortex, ATP usage can be seen as a decay of the FRET signal during visual stimulation with natural stimuli (*Padamsey et al., 2022*; *Figure 3A*). Data obtained in females were compared to our previously published datasets obtained under similar conditions in male mice (*Padamsey et al., 2022*). As previously shown (*Padamsey et al., 2022*), food restriction robustly decreased the rate of cortical ATP usage in males (24% decrease) (half-time of FRET decay: CTR male vs. FR male; 8.48 min [95% CI: 7.59–9.37 min] vs. 10.51 min [95% CI: 9.08–11.93 min]; *t* = 2.87; df = 37; p = 0. 0067; *n* = 11 CTR males and 10 FR males) during visual stimulation; a more modest trend was observed in females (12% decrease), which was not significant compared to controls (half-time of FRET decay: CTR female vs. FR female; 8.66 min [95% CI: 7.49–9.84 min] vs. 9.66 min [95% CI: 8.92–10.39 min]; *t* = 1.36; df = 37; p = 0.18; *n* = 12 CTR females and *n* = 8 FR females) (*Figure 3B, C*). In the absence of visual stimulation (darkness), ATP usage was lower than during visual stimulation and unaffected by food restriction, for either sex (*Figure 3—figure supplement 1*).

Altogether, these findings demonstrate that food restriction significantly impacts V1 energy usage during visual stimulation in males but not in females.

## Orientation selectivity in V1 is significantly reduced under food restriction in males but not in females

We next examined how food restriction impacts visual cortical function. In male mice, reductions in energy usage in V1 during food restriction are associated with a decreased visual coding precision as evidenced by a reduction of orientation selectivity (*Padamsey et al., 2022*). Given the non-significant impact of food restriction on energy usage in female mice, we reasoned that orientation selectivity would also be less impacted. We analysed two-photon calcium imaging data of V1 GCaMP6s-labelled layer 2/3 neurons in female mice viewing drifting gratings (*Figure 3D–F*; *n* = 7 CTR male, *n* = 8 FR male, *n* = 7 CTR female, and *n* = 9 FR female mice). Data obtained in females were compared to our previously published datasets obtained under similar conditions in male mice (*Padamsey et al., 2022*). Consistent with our hypothesis, in contrast to males in which orientation selectivity was significantly reduced by 27% (orientation selectivity index [OSI]: CTR male vs. FR male; 0.58 [95% CI: 0.51–0.65] vs. 0.42 [95% CI: 0.36–0.49]; *t* = 3.75; df = 27; p = 0.0009), orientation selectivity was modestly (13%) and not significantly reduced by food restriction in females (OSI: CTR female vs. FR female; 0.60 [95% CI: 0.52–0.67] vs. 0.52 [95% CI: 0.45–0.58]; *t* = 1.90; df = 27; p = 0.08) (*Figure 3E, F*). Direction tuning, as measured by the direction selectivity index (DSI) was also significantly affected by food restriction in males, but not in females (*Figure 3—figure supplement 3*; males: CTR vs. FR; *t* = 2.16; p = 0.023; df = 12; *n* = 6 CTR males and 8 FR males; females: CTR vs. FR; *t* = 1.49; p = 0.16; df = 14; *n* = 7 CTR females and 9 CTR females).

Altogether, our results show that energy usage and coding precision in the visual cortex are largely maintained in female mice under food restriction while they are robustly reduced in males. Such findings cannot be readily attributed to differences in behavioural or attentional state. All animals were imaged while in a cardboard tube, and therefore not locomoting. Moreover, we found no difference in pupil size during the presentation of visual stimuli (*Figure 3—figure supplement 2*).

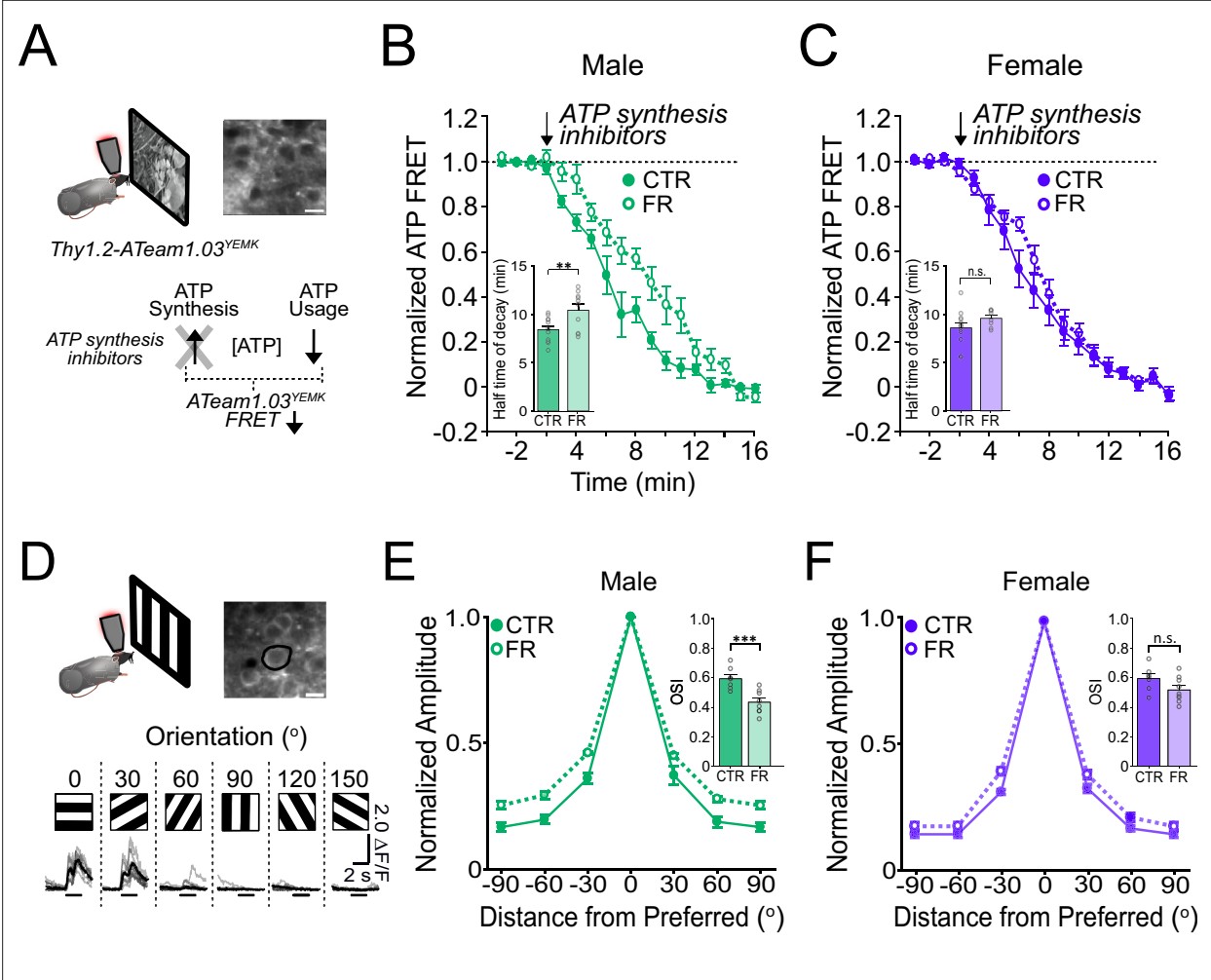

**Figure 3.** ATP usage and orientation selectivity in V1 are more robustly decreased by food restriction in males than in females. (**A**) Schemata of ATP imaging experiment. Top: Example field of view of V1 layer 2/3 neurons in the ATeam1.03$^{YEMK}$ transgenic mouse (scale bar: 10 μm). Bottom: ATP synthesis inhibitors were used to isolate ATP usage, recorded as a decrease in FRET signal during presentation of natural, outdoor scenes. (**B**) Normalized ATeam1.03$^{YEMK}$ FRET signal in males and during presentation of natural, outdoor scenes. ATP synthesis inhibitors (arrow) were added to isolate ATP usage. Inset: Time to 50% decay of the ATeam1.03$^{YEMK}$ FRET signal (two-way analysis of variance [ANOVA]: $t = 2.87$; df = 37; $p = 0.0067$; $n = 11$ CTR males and 10 FR males). (**C**) As in (**B**) but for females (inset: two-way ANOVA: $t = 1.36$; df = 37; $p = 0.18$; $n = 12$ CTR females and $n = 8$ FR females). (**D**) Top: Schema of two-photon imaging and sample field of view of V1 layer 2/3 neurons expressing GCaMP6s (scale bar: 10 μm). Bottom: Sample fluorescent signals (grey) from a selected neuron (black circle) in response to 6 drifting gratings of varying orientations. Trial averages are in black. Horizontal bar denotes 2 s grating presentation. (**E**) Mean orientation tuning curves normalized to the response to the preferred orientation for CTR and FR males (data from ***Padamsey et al., 2022***). Note that −90° and +90° conditions correspond to the same visual stimulus. Inset: Orientation selectivity index (OSI; two-way ANOVA: $t = 3.75$; df = 27; $p = 0.0009$; $n = 8$ CTR and 8 FR males). (**F**) As in (**E**) but for females (Inset: two-way ANOVA: $t = 1.90$; df = 27; $p = 0.08$; $n = 7$ CTR females and 9 FR females). \*\*$p < 0.01$; \*\*\*$p < 0.001$; n.s. = not significant. Error bars are standard error of the mean (SEM). Data from male mice are from a previously published dataset obtained under similar conditions (***Padamsey et al., 2022***).

The online version of this article includes the following figure supplement(s) for figure 3:

**Figure supplement 1.** ATP usage in visual cortex is not affected by food restriction in the absence of visual stimulation.

**Figure supplement 2.** Pupil diameter during visual stimulation is not affected by sex or food restriction.

**Figure supplement 3.** Bayes factor analysis reveals statistically robust impact of metabolic stress on V1 energy usage and visual coding in males, but not in females.

## Bayes factor analysis reveals statistically robust impact of metabolic stress on V1 energy usage and visual coding in males, but not females

We used Bayes factor hypothesis testing to better assess the statistical impact of food restriction on visual cortical function and energy usage in male and female mice (***Figure 3—figure supplement 3***).

Bayes factor analysis uses a Bayesian framework to quantify how much more likely the data is under the alternative hypothesis (i.e. there is an effect of food restriction) vs. the null hypothesis (i.e. there is no effect of food restriction), quantified by the Bayes factor ($BF_{10}$) (*van Doorn et al., 2021*; *Keysers et al., 2020*). A $BF_{10}$ >3, for example, means that the data are >3 times more likely under the alternative hypothesis than the null hypothesis. Critically, the $BF_{10}$ allows for a more informative comparison of the effects of food restriction between sexes, as opposed to comparing p values, which can be misleading (*van Doorn et al., 2021*; *Keysers et al., 2020*). Across the parameters of energy usage and visual coding that we tested (*Figures 1–3*), $BF_{10}$ for males was consistently several-fold greater than that for females. Thus, visual cortical function and energy usage were more robustly modulated by food restriction in males than in females.

Our findings however cannot rule out a potential, smaller impact of food restriction on cortical function and energy usage in females, below the threshold for statistical detection in this study. In particular, for females, we found only 3 out of the 10 parameters had $BF_{10}$ <0.33, that is the data are three times more likely under the null hypothesis than the alternative hypothesis supporting the absence of an effect (*van Doorn et al., 2021*; *Keysers et al., 2020*; *Figure 3—figure supplement 3*). This suggests that food restriction may have a modest impact on V1 energy usage or function in females, but below the limits of statistical detection in our study.

Altogether, our results show that energy usage and coding precision in the visual cortex are largely maintained in female mice under food restriction while they are robustly reduced in males.

## Discussion

Here, we find that in female mice, energy usage and visual coding in V1 are largely unchanged under food restriction. Using molecular analysis and RNA sequencing, we found that energy-regulating pathways were not significantly changed with food restriction in females, in contrast to males; these included pathways associated with AMPK, mTOR, and PPARα signalling, as well as oxidative phosphorylation. Consistent with this, we found that food restriction in females did not impact visual cortical ATP usage nor orientation selectivity, both of which were reduced by food restriction in males. Our findings suggest that the contribution of neocortex to energy-saving adaptations during food restriction is sex selective and more pronounced in males; females may rely on other energy-saving strategies, such as reducing reproductive function, to cope with food restriction.

### Limitations of the study

Our study focussed on the effects of a moderate food restriction protocol over 2–3 weeks on neocortical function and energy usage. This protocol is widely used in metabolic studies investigating the impact of caloric restriction on peripheral physiology, as well as in neuroscience studies to motivate rodents to perform specific behaviours (*Suchacki et al., 2023*; *Guo et al., 2014*; *Goltstein et al., 2018*; *Toth and Gardiner, 2000*). However, it remains unclear to what extent our findings generalize to other protocols of food restriction. Indeed, the duration, magnitude, and pattern of food restriction, as well as macronutrient composition are known to impact the physiological responses to dietary manipulations (*Dagon et al., 2005*; *Collet et al., 2017*; *Cuevas-Cervera et al., 2022*; *Hofer et al., 2022*). Moreover, while we focussed our study on the adult visual cortex, it remains unclear if the brain shows similar sex-specific energy-saving adaptations in other cortical areas, and how such adaptations change with age (*Suchacki et al., 2023*; *Mitchell et al., 2016*).

A key strength of our study is that we consistently found, across a number of experimental methods, that changes in cortical function and energy usage were significant in males, but not in females. However, a critical limitation of our study is that we did not have the statistical power to directly compare the effect size of food restriction between the sexes (e.g. analysis of variance [ANOVA] interaction effects); to do so would require a substantial increase in experimental group sizes by several-fold. We addressed this limitation, in part, using Bayes factor analysis, which revealed that the impact of food restriction in males was indeed more robust than in females. This analysis also revealed that our study did not have sufficiently strong evidence to definitively rule out a smaller effect of food restriction in females, which may not have been statistically detectable.

## Role of leptin and sex hormones in food restriction impact on cortical activity

Consistent with past studies, we found that food-restricted female mice resisted weight loss, and therefore required a greater restriction of food to achieve similar weight loss (relative to baseline) as males. In addition, for the same relative weight loss as males, females exhibited minimal reductions in serum leptin levels (secreted by adipocytes), consistent with past studies showing that females resist fat loss during food restriction (*Andersson et al., 1991*; *Cortright and Koves, 2000*; *Volek et al., 2004*; *Tirosh et al., 2015*; *Suchacki et al., 2023*). Similar effects have been observed across several mammalian species, including humans, and is thought to reflect the differential importance of fat stores for reproductive success in males and females (*Andersson et al., 1991*; *Cortright and Koves, 2000*; *Volek et al., 2004*; *McCarthy and Berg, 2021*; *Tirosh et al., 2015*). Notably, we previously demonstrated that reductions in leptin levels during food restriction are necessary for driving energy-saving changes in visual cortical coding in male mice; leptin therefore acts as a critical signal linking the neocortex to fat stores (*Padamsey et al., 2022*). That leptin levels are not significantly affected by food restriction in females may explain the minimal impact food restriction has on cortical energy usage and coding precision in female mice.

In addition to leptin, sex hormones likely contribute to sex-specific adaptations to food restriction (*Kane et al., 2018*). As with leptin, oestrogen suppresses food intake and promotes energy expenditure (*Brown and Clegg, 2010*), in part by augmenting leptin signalling by increasing leptin-induced phosphorylation of STAT3 (*Gao and Horvath, 2008*). Moreover, oestrogen can enhance ATP production by sensitizing insulin signalling, enhancing glycolysis, and oxidative phosphorylation, and along with progesterone, upregulating mitochondrial gene expression and function (*Brown and Clegg, 2010*; *Irwin et al., 2008*; *Brinton, 2008*). Interestingly, in our study, RNA sequencing revealed that oestrogen signalling pathways were significantly regulated in male visual cortex with food restriction, and were unchanged in females. Oestrogen may thus have overlapping effects with those induced by food restriction, thereby diminishing further responses to food restriction in females (*Suchacki et al., 2023*). Notably, it was recently shown that sex differences in calorie-restriction's metabolic effects that are observed in young adult mice, are largely absent in older mice, when females' oestrogen levels have declined (*Suchacki et al., 2023*). Further studies will be needed to explore whether the resilience of cortical function found in this study in young cycling adult females is also observed in non-cycling mice with reduced oestrogen levels, such as in aged mice.

## Sex-specific regulation of cellular energy-regulating pathways

AMPK and mTOR signalling pathways are canonical regulators of cellular energy usage (*Garza-Lombó et al., 2018*; *Takei and Nawa, 2014*). We found these pathways, along with cortical ATP usage, to be significantly modulated by food restriction in male, but not female mice. AMPK is activated during metabolic stress, and saves energy by regulating oxidative phosphorylation (*Herzig and Shaw, 2018*; *de la Cruz López et al., 2019*), and inhibiting cellular pathways associated with energy usage, such as mTOR signalling (*Garza-Lombó et al., 2018*). mTOR signalling, by upregulating protein synthesis, increases ATP usage by promoting neuronal excitability, neurite outgrowth, ion channel expression, and synaptic plasticity (*Takei and Nawa, 2014*). mTOR signalling is therefore ideally situated to regulate how much ATP is used in neuronal function; its inhibition may underlie ATP savings and the associated loss of coding precision with food restriction in male mice. Notably, previous studies have found sex- and tissue-dependent regulation of AMPK and mTOR signalling, including in response to food restriction (*Kane et al., 2018*). For example, in skeletal muscle, mTOR signalling is enhanced by fasting selectively in male, but not female mice (*Kane et al., 2018*; *Baar et al., 2016*). Moreover, metabolic challenge by endurance training induces greater AMPK activity in cardiac tissue in male, as opposed to female mice (*Brown et al., 2020*). Our study extends the sex-specific regulation of these energy-regulating pathways to the neocortex.

## Importance of biological sex on the impact of food restriction on cortical function

Food restriction protocols are widely used to motivate animals in behavioural studies examining cortical function (*Guo et al., 2014*; *Goltstein et al., 2018*; *Toth and Gardiner, 2000*). Our findings suggest that these protocols are likely to have greater impacts on basal neocortical function in males,

as compared to females. Consistent with this, a recent study has found that food restriction in juvenile mice strongly impacts cognitive flexibility in learning and decision-making in adulthood, specifically for males but not females (*Clemens et al., 2019*; *Lin et al., 2022*). Such sex-specific impacts of food restriction on neocortical function are important to take into consideration in the design and interpretation of such studies.

Diet is known to have a considerable impact on neurological diseases, including epilepsy, Alzheimer's disease, and other neurodegenerative diseases (*de Carvalho, 2022*; *Schroeder et al., 2010*). While calorie restricted diets have proven to be of benefit in this regard, the sex dependency of such interventions have not been thoroughly investigated, though on the basis of our findings it is conceivable that males and females may respond differently to dietary interventions. The inclusion of both males and females in biomedical research, and the study of sex differences, is therefore of considerable importance for ensuring efficacious treatments are developed for both sexes.

In conclusion, we find that visual cortical coding and energy usage in females are resistant to food restriction, in contrast to males. Our findings establish that sex-specific adaptations in energy usage, previously described in peripheral tissue under metabolic challenge, extends to the neocortex.

## Materials and methods
### Resource availability
#### Lead contact
Further information and requests for resources and materials should be directed to and will be fulfilled by the lead contact Nathalie L. Rochefort (n.rochefort@ed.ac.uk).

### Materials availability
This study did not generate new unique reagents.

### Data and code availability

1. GEO accession number for RNA-seq datasets: GSE233435.
2. Processed data will be made available at https://datashare.ed.ac.uk/handle/10283/3871. Requests for raw data should be made to and will be fulfilled by the lead contact (n.rochefort@ed.ac.uk).
3. MATLAB scripts to analyse data have been previously published (*Padamsey et al., 2022*) and are available at https://zenodo.org/record/5561795#.YmJ3AtrMJPY and https://github.com/rochefort-lab/Padamsey-et-al-Neuron-2022.
4. Any additional information required to reanalyse the data reported in this paper is available from the lead contact upon request.

### Experimental model and subject details
Experiments were approved by the University of Edinburgh's Animal Welfare and Ethical Review Board (AWERB) and carried out under Home Office (UK) approved project and personal licenses. All experiments conformed to the UK Animals (Scientific Procedures) Act 1986 and the European Directive 86/609/EEC and 2010/63/EU on the protection of animals used for experimental purposes.

This study used male and female C57BL/6J mice (RRID:IMSR_JAX:000664; Jackson Laboratory) and male B6-Tg (Thy1.2-ATeam1.03$^{YEMK)AJhi}$ transgenic mice, that were bred on a C57BL/6J background (RRID:MGI:5882597; https://scicrunch.org/resources). Animals were group housed (2–5 animals/cage) and maintained on a reverse light/dark (12/12 hr) cycle in a room kept at 21 ± 2°C and 55 ± 10% humidity.

For data associated with in vivo recordings (*Figure 3*), data obtained in female mice were compared to our previously published datasets obtained under similar conditions in male mice (*Padamsey et al., 2022*).

## Method details

### Food restriction

Mice (7–9 weeks of age) either had ad libitum access to food (RM1 expanded pellets; DBM Scotland UK) or were food restricted for 2–3 weeks to 85% of their baseline bodyweight prior to experimentation as previously described (*Padamsey et al., 2022*). Briefly, for food restriction, one ration of food was given 4–8 hr prior to the end of the dark cycle, with the amount of food progressively reduced to achieve target weight. 1–2 hr prior to experimentation, all animals had ad libitum access to food until sated.

### AAV injection and cranial window

For calcium imaging experiments, C57BL/6 were anesthetized using isoflurane, and given pre-operative analgesia (vetergesic: 0.1 mg/kg; carprieve: 5 mg/kg; rapidexon: 2 mg/kg) along with warm Ringer's solution (25 ml/kg) subcutaneously; vetergesic jelly was additionally given orally post-operatively 24 hr later. Opaque eye cream was applied to the eyes (Bepanthen, Bayer, Germany). A craniotomy (2 × 2 mm) was applied over left V1 (centre at 2.5 mm mediolateral and 0.5 mm anterior to lambda). Flexed GCaMP6s (AAV1.Syn.Flex.GCaMP6s.WPRE.SV40; Addgene; RRID:Addgene_105558-AAV1) diluted 1:10 in saline, along with CaMKII-dependent Cre-recombinase (AAV1.CamKII 0.4.Cre.SV40; Addgene; RRID:Addgene_105558-AAV1) diluted 1:100 saline was injected at the centre of the craniotomy via a sharp glass pipette. Injections were targeted to layer 2/3, with 100 nl of virus solution injected (2.5 nl/30 s) at each of three depths (150, 250, and 350 μm) via a sharp glass pipette using a Nanoject III (Drummond Scientific). A glass window was then used to seal the craniotomy, and superglued in place. The window was made of one or two glass coverslips (Menzel-Glaser # 0), in the latter case the two cover slips were glued together with optically clear, UV-cured glue (Norland Optical Adhesive no. 60); the inner window was 2.0 mm × 2.0 mm, the outer window was 2.5 mm × 2.5 mm. A metal headplate was then fixed to the skull with super glue and dental cement (Paladur, Heraeus Kuzler). It took 3–6 weeks for viral expression prior to experimentation.

For ATP imaging experiments, we used ATeam1.03$^{YEMK}$ mice (RRID:MGI:5882597; https://scicrunch.org/resources) that express a FRET-based sensor under the Thy1 promoter (*Trevisiol et al., 2017*). One the day of experimentation, animals were anesthetized with isoflurane and given pre-operative analgesia (Carprieve: 5 mg/kg) and 25 ml/kg of warm Ringer's solution subcutaneously. A headplate was first fixed to the skull with superglue and dental cement, after which a small craniotomy (~0.5 mm × ~0.5 mm) was made over left V1 (centre at 2.5 mm mediolateral and 0.5 mm anterior to lambda), which was covered with agarose (1–2%) and silicone. Following recovery from anaesthesia (30–60 min) the animal was head fixed; the agarose and silicone were removed and replaced with HEPES-buffered Artificial cerebrospinal fluid, (ACSF) (in mM: 124 NaCl, 20 glucose, 10 HEPES, 2.5 KCl, 1.2 NaH$_2$PO$_4$, 2 CaCl$_2$, and 1 CaCl$_2$; pH 7.2–7.4).

### Habituation and head fixation

During imaging, mice were placed in a cardboard tube and head fixed. Prior to the imaging day, mice were handled daily for 2–3 weeks prior, and exposed to cardboard tubes, similar to the one on the experimental rig. They were also habituated to the rig via once daily, 10- to 15-min sessions for 2 days prior to recording. For ATP imaging, mice were habituated to the experimental setup without head fixation (10–15 min session/day for 1–2 days).

### In vivo two-photon imaging

ATP and calcium imaging were using a two-photon resonant scanning microscope, as previously described (*Pakan et al., 2016*; *Pakan et al., 2018*; *Henschke et al., 2020*), which was equipped with a Ti:Sapphire laser (Charmeleon Vision-S, Coherent, CA) and GaAsP photomultiplier tubes (Scientifica), and controlled with LabView (v8.2; National Instruments, UK). Time-series *xy* images were acquired at 40 Hz using a ×25 water-immersion objective (Nikon; CF175 Apo 25XC W; 1.1 NA) at depths between 160 and 280 μm below the pia. For GCaMP6s imaging, excitation was tuned to 920 nm. For ATP imaging (CFP/YFP FRET sensor), the laser was tuned to 850 nm for CFP excitation. CFP and YFP fluorescence were recorded simultaneously using a 515-nm long-pass dichroic mirror with the following emission filters: 485/70 nm for CFP and 535/45 nm for YFP (mirror and filter set: T515lpxr C156624; Scientifica). Imaging data were acquired during the presentation of visual stimuli, with at least 10

replicate trials for each visual stimulus. For ATP imaging, three trials were taken at baseline, then ACSF containing ATP synthesis inhibitors (1 mM oligomycin and 20 mM sodium iodoacetate) was applied over the open craniotomy isolate ATP usage. Thirty trials were successively performed immediately after drug application. For in vivo recordings, data obtained in females were compared to our previously published datasets obtained under similar conditions in male mice (*Padamsey et al., 2022*).

## Visual stimuli

MATLAB (Mathworks; RRID:SCR_001622; Psychophysics Toolbox; RRID:SCR_002881) was used to generate drifting grating stimuli. These were displayed on an LCD monitor (51 × 29 cm; Dell) placed 20 cm from the right eye, contralateral to the hemisphere with the cranial window. For food restriction experiments, on a given trial, 12 full-field drifting gratings (drift angle: 0, 30, 60, 90, 120, 180, 210, 240, 270, 300, and 330°; temporal frequency = 1 Hz), 2 s in duration, were presented in random order, interspersed with grey screens (4 s duration). Gratings within a trial were randomly assigned one common spatial frequency (0.02, 0.04, 0.16, or 0.32 cpd). For natural stimuli, a 60-s movie was presented of an outdoor movie of movement through a field and vegetation. For dark conditions, no visual stimulus was presented (screen off) and all sources of light were switched off.

## Pupil measurements

In some experiments the pupil was monitored using a USB camera (USB 2.0 monochrome camera; ImagingSource) with frames captured at 30 Hz.

## Serum leptin measurements

Serum leptin levels were analysed using ELISA kits (Mouse Leptin Quantikine Elisa Kit, R&D Systems, USA) according to the manufacturer's instruction. For serum collection, trunk blood was collected from mice that were briefly anesthetized (isoflurane) prior to decapitation. Blood was allowed to clot at room temperature for 1–2 hr and then centrifuged for 20 min at 2000 × *g*. The serum was drawn off and stored at −80°C until used.

## AMPK and PPARα measurements

Levels of total AMPK, phosphorylated AMPK (Thr172), and nuclear PPARα in V1 were measured using ELISA kits according to the manufacturer's instructions (total AMPK: ab181422, Abcam, UK; phosphorylated AMPK: KHO0651, Thermo Fisher, UK; PPARα: ab133107, Abcam, UK). For tissue collection, mice were briefly anesthetized (isoflurane) and decapitated. The visual cortex was dissected out, snap frozen using dry ice, and stored at −80°C until used.

## RNA sequencing

Bulk RNA sequencing was performed on V1. V1 tissue was collected and snap frozen from mice that were briefly anesthetized (isoflurane) and decapitated. Tissue was homogenized using a bead mill (Bead Mill 24, Fisher Scientific) and RNA was isolated using an RNeasy kit (Qiagen) according to the manufacturer's instructions. RNA integrity was confirmed (RIN >7) and mRNA was sequenced at the Oxford Genomics Centre using Illumina NovaSeq 6000.

## Quantification and statistical analysis

### Ca$^{2+}$ imaging analysis

Image analysis for calcium imaging was performed as previously described (*Baar et al., 2016*; *Clemens et al., 2019*; *de Carvalho, 2022*). For motion correct, a discrete Fourier 2D-based image alignment was used (SIMA 1.3.2) (*Henschke et al., 2020*). Regions of interest (ROI) were manually drawn around imaged neuronal soma using ImageJ software (NIH public domain; RRID:SCR_002285). Pixel fluorescence was then averaged within each ROI to generate a time series. For each ROI, baseline fluorescence ($F_0$) was calculated by taking the 5th percentile of the smoothed time series (1 Hz lowpass, zero-phase, 60th-order Finite Impulse Response (FIR) filter). $\Delta F/F$ was then calculated as ($F - F_0/F_0$). The toolbox FISSA (Fast Image Signal Separation Analysis) was used to decontaminate the neuropil (*Keemink et al., 2018*). Subsequent analyses were performed using custom scripts in MATLAB (MathWorks) (*Padamsey et al., 2022*).

## Drifting grating analysis

The visual response to drifting gratings was defined as the highest mean $\Delta F/F$, averaged within a 2-s window, that occurred within a 4-s window (comprising the 2 s grating presentation plus the 2 s of the grey screen presentation which immediately followed the grating presentation). The visual response was subtracted by baseline $\Delta F/F$, defined as the mean value within a 1-s window prior to the visual response.

Responses to gratings of the same angle, but different drift directions were averaged together. Given that multiple spatial frequencies were used, only the spatial frequency with the largest response, meaned across all orientations, was selected for each neuron for subsequent analyses. A neuron's preferred orientation was that which evoked the largest mean response.

Grating-responsive neurons were defined as those for which grating responses were better fit with a double Gaussian curve (direction responses) than with a flat line at zero (null model), which was determined using Bayesian information criterion (BIC) (**Padamsey et al., 2022**). The double Gaussian curve was defined as:

$$R\left(\theta\right) = C + R_p e^{\dfrac{-ang_{dir}\left(\theta - \theta_{pref}\right)^2}{2\sigma^2}} + R_n e^{\dfrac{-ang_{dir}\left(\theta - \theta_{pref}\right)^2}{2\sigma^2}}$$

where $R(\theta)$ is the response at a given direction angle $\theta$, $C$ is an offset, $R_p$ is the response to the preferred direction after subtracting the offset, $\theta_{pref}$ is the angle of the preferred direction, $R_n$ is the response to the null direction after subtracting the offset, and $ang_{dir}\left(x\right) = \min\left(x, x - 360, x + 360\right)$, which constrains angular differences to 0–180°, and $\sigma$ is the standard deviation of the curve.

BIC was given by:

$$BIC = n\ln(\overline{\sigma^2}) + k\ln\left(n\right)$$

where $n$ is the number of responses, $\overline{\sigma^2}$ is the mean residual sum of squares of the model, and $k$ is the number of free parameters used by the model, which was zero in the case of the null model. A neuron was considered significantly grating-responsive if the $BIC_{null} - BIC_{Gaussian} \geq 10$, which provides strong evidence against the null model (**Kass and Raftery, 1995**).

## Orientation selectivity

The OSI of grating-responsive neurons was calculated as:

$$OSI = 1 - CirVar = abs\left(\dfrac{\sum_k R\left(\theta_k\right) e^{2i\theta_k}}{\sum_k R\left(\theta_k\right)}\right)$$

where *Cirvar* is the circular variance and $R(\theta_k)$ is the mean response to the $k_{th}$ angle $\theta$ in orientation space, averaged across direction (see detailed description in **Mazurek et al., 2014**). Mean responses less than zero were set to zero. We used the median OSI values across neurons within an animal.

## Direction selectivity

The DSI of grating-responsive neurons was calculated as:

$$DSI = 1 - dirCirVar = abs\left(\dfrac{\sum_k R\left(\theta_k\right) e^{i\theta_k}}{\sum_k R\left(\theta_k\right)}\right)$$

where *dirCirVar* is the circular variance in direction space and $R(\theta_k)$ is the mean response to the $k_{th}$ angle $\theta$ in direction space (see detailed description in **Mazurek et al., 2014**). Mean responses less than zero were set to zero. We used the median DSI values across neurons within an animal.

## Pupil analysis

Pupil diameter was quantified with custom scripts in MATLAB using functions from the Imaging Processing Toolbox, namely:

1. *imresize* to resize the video
2. *medfilt2* and *imadjust* to remove noise and adjust contrast
3. *imbinarize*, *imclearborder*, and *bwpropfilt* to find the pupil
4. *regionprops* to fit an ellipsoid around the pupil

The pupil diameter (*d*) was calculated from the fitted ellipsoid as:

$$d = 2\sqrt{\text{semi-major axis} * \text{semi-minor axis}}$$

## ATP measurements

For ATeam1.03$^{\text{YEMK}}$ imaging, CFP and YFP fluorescence was averaged across ROIs drawn around visible soma in the field of view, after background subtraction. FRET was calculated as a ratio of YFP/CFP fluorescence, and decayed as a function of time with visual stimulation in the presence of ATP synthesis inhibitors. The FRET signal was bound between 0 and 1 by first subtracting the mean FRET signal during the last three imaging trials (the FRET signal plateaued at this point), and then dividing by the mean FRET signal at baseline. The FRET decay half time was defined as the time required for the FRET to decay to 50% of its baseline value.

## Differential gene expression analysis

RNA sequencing reads were mapped, using STAR 2.4.oi, to the *Mus musculus* primary assembly (Ensembl v80) (***Dobin et al., 2013***). FeatureCounts 1.4.6-p2 was used to count reads uniquely aligned to annotated genes (***Liao et al., 2014***). Differential gene expression analysis was then conducted using DESeq2 1.12.4 (betaPrior = False) to normalize counts (***Love et al., 2014***). For each gene, a fold change in gene expression was defined as $log_2 \frac{\text{Normalized count under food restriction (FR)}}{\text{Normalized count under controlconditions (CTR)}}$ for males and females. GSEA was conducted using GSEA software (https://www.gsea-msigdb.org/gsea/index.jsp) using the Hallmarks Gene Sets (v7.5.1).

## Statistics

Statistical tests, p values, and the number and definition of independent units are stated in the figure legends. For all measures, the number of animals was taken as the statically independent, biological unit. All data came from at least two independent cohort of animals. All animals were randomly assigned to experimental groups. Sample size was determined by a priori based on calculations, with an aim of achieving 80% power on the basis of a 20% group difference and a significance threshold of p < 0.05. Group means and variances were taken from pilot data or previously published data (***Padamsey et al., 2022***). Where applicable, ANOVA was used to assess significance, followed by post hoc Fisher LSD tests to prevent loss of power. Statistical tests were carried out in Prism 6 (GraphPad Prism; RRID:SCR_002798). Averages denoted in figures represent means, with error bars representing the standard error of the means. We complemented our statistical analyses using Bayes factor hypothesis testing (***Figure 3—figure supplement 3***) as described in ***van Doorn et al., 2021***; ***Keysers et al., 2020*** using JASP software to calculates Bayes factor, effect size, and CIs (https://jasp-stats.org/). As recommended, we used a default, Cauchy distribution with a spread of $1/\sqrt{2}$ as a prior distribution of effect size. A BF10 <0.33 is considered to be moderate evidence for the absence of an effect (***van Doorn et al., 2021***; ***Keysers et al., 2020***).

## Inclusion and diversity

One or more of the authors of this paper self-identifies as an underrepresented ethnic minority in science. One or more of the authors of this paper self-identifies as a member of the LGBTQ + community. While citing references scientifically relevant for this work, we also actively worked to promote gender balance in our reference list.

# Acknowledgements

We thank the GENIE Program and the Janelia Research Campus (V Jayaraman, R Kerr, D Kim, L Looger, and K Svoboda) for making GCaMP6 available. We thank J Hirrlinger (Carl Ludwig Institute of Physiology, University of Leipzig, Germany) for providing the ATeam1.03$^{YEMK}$ mouse line. We thank Sang Seo and Aditi Singh for assistance and guidance in RNA sequencing and analysis. We thank Will Cawthorn for discussions on the manuscript. This work was funded by the BBSRC Responsive Mode Research Grant (BB/T007907/1 to N.R. and ZP), the Royal Commission for the Exhibition of 1851 (research fellowship to ZP), the MRC (G116854; Career Development Fellowship to ZP), the Wellcome Trust and the Royal Society (Sir Henry Dale fellowship to NR), the Shirley Foundation, the Patrick Wild Centre, the RS MacDonald Charitable Trust Seedcorn Grants (to NR and ZP), the Simons Initiative for the Developing Brain (to NR and DK), and the Wellcome Trust-University of Edinburgh Institutional Strategic Support Fund (ISSF3) (to NR). This project has received funding from the European Research Council (ERC) under the European Union's Horizon 2020 research and innovation programme (grant agreement No. 866386).

## Additional information

### Funding

| Funder | Grant reference number | Author |
|---|---|---|
| Biotechnology and Biological Sciences Research Council | BB/T007907/1 | Zahid Padamsey Nathalie L Rochefort |
| Royal Commission for the Exhibition of 1851 | Zahid Padamsey | Zahid Padamsey |
| European Research Council | 866386 | Nathalie L Rochefort |
| Wellcome Trust | 10.35802/102857 | Nathalie L Rochefort |
| Simons Initiative for the Developing Brain | Danai Katsanevaki | Danai Katsanevaki Nathalie L Rochefort |
| Wellcome Trust-University of Edinburgh Institutional Strategic Support Fund | ISSF3 N. Rochefort | Nathalie L Rochefort |
| the RS MacDonald Charitable Trust Seedcorn Grants | | Zahid Padamsey |
| the Shirley Foundation | | Nathalie L Rochefort |
| the Patrick Wild Centre | | Nathalie L Rochefort |
| MRC (Career Development Fellowship) | | Zahid Padamsey |

The funders had no role in study design, data collection, and interpretation, or the decision to submit the work for publication. For the purpose of Open Access, the authors have applied a CC BY public copyright license to any Author Accepted Manuscript version arising from this submission.

### Author contributions

Zahid Padamsey, Conceptualization, Data curation, Formal analysis, Funding acquisition, Validation, Investigation, Visualization, Methodology, Writing – original draft, Writing – review and editing, designed the experiments.performed and analysed calcium imaging experiments and ELISAs. processed tissue for RNA sequencing and analyzed data. wrote the manuscript with input from all authors; Danai Katsanevaki, Data curation, Formal analysis, Validation, Investigation, Visualization, Methodology, Writing – review and editing, performed and analysed calcium and ATP imaging experiments; Patricia Maeso, Data curation, Investigation, performed and analyzed ELISAs; Manuela Rizzi, Data curation, Formal analysis, Investigation, Methodology, processed tissue for RNA sequencing and analyzed data; Emily E Osterweil, Resources, Supervision, Validation, Methodology, Writing – review

and editing, Supervised RNA sequencing and data analysis; Nathalie L Rochefort, Conceptualization, Resources, Data curation, Software, Supervision, Funding acquisition, Validation, Methodology, Writing – original draft, Project administration, Writing – review and editing, designed the experiments. supervised the project. wrote the manuscript with input from all authors

### Author ORCIDs
Zahid Padamsey (ID) https://orcid.org/0000-0001-9177-8210
Nathalie L Rochefort (ID) https://orcid.org/0000-0002-3498-6221

### Ethics
Experiments were approved by the University of Edinburgh's Animal Welfare and Ethical Review Board (AWERB) and carried out under Home Office (UK) approved project and personal licenses. All experiments conformed to the UK Animals (Scientific Procedures) Act 1986 and the European Directive 86/609/EEC and 2010/63/EU on the protection of animals used for experimental purposes.

Reviewer #1 (Public Review): https://doi.org/10.7554/eLife.93052.3.sa1
Reviewer #2 (Public Review): https://doi.org/10.7554/eLife.93052.3.sa2
Reviewer #3 (Public Review): https://doi.org/10.7554/eLife.93052.3.sa3
Author response https://doi.org/10.7554/eLife.93052.3.sa4

## Additional files

### Supplementary files
• Supplementary file 1. Hallmark Gene Sets, male food deprived vs. controls. Gene Set Enrichment Analysis (GSEA) to identify pathways enriched in the populations differentially expressed in food restricted animals compared to controls. We found that food restriction resulted in a significant alteration of 34 gene sets (at p-adj <0.05) in males; 13 of these were common to both sexes. $n = 4$ CTR male, $n = 4$ FR male mice.

• Supplementary file 2. Hallmark Gene Sets, female food deprived vs. controls. Same as *Supplementary file 1* for females. We found that food restriction resulted in a significant alteration of 14 gene sets (at p-adj <0.05) in females; 13 of these were common to both sexes. $n = 4$ CTR female, $n = 4$ FR female mice.

• MDAR checklist

### Data availability
GEO accession number for RNA-seq datasets: GSE233435- MATLAB scripts to analyse data have been previously published and are available at https://zenodo.org/record/5561795#.YmJ3AtrMJPY and https://github.com/rochefort-lab/Padamsey-et-al-Neuron-2022 (copy archived at *rochefort-lab, 2022*).

The following dataset was generated:

| Author(s) | Year | Dataset title | Dataset URL | Database and Identifier |
|---|---|---|---|---|
| Padamsey Z, Katsanevaki D, Maeso P, Rizzi M, Osterweil E, Rochefort NL | 2023 | Sex-specific resilience of neocortex to food restriction | https://www.ncbi.nlm.nih.gov/geo/query/acc.cgi?acc=GSE233435 | NCBI Gene Expression Omnibus, GSE233435 |

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
