## [Editor Report · eLife assessment]

This study provides **important** findings based on **compelling** evidence demonstrating that females and males have different strategies to regulate energy consumption in the brain in the context of low energy intake. While food deprivation reduces energy consumption and visual processing performance in the visual cortex of males, the female cortex is unaffected, likely at the expense of other functions. This study is relevant for scientists interested in body metabolism and neuroscience.

---

## [Referee Report · Reviewer #1 (Public Review)]

Padamsey et al. followed up on their previous study in which they found that male mice sacrifice visual cortex computation precision to save energy in periods of food restriction (Padamsey et al. 2021, Neuron). In the present study, the authors find that female mice show much lower levels of adaptation in response to food restriction on the level of metabolic signaling and visual cortex computation. This is an important finding for understanding sex differences in adaptation to food scarcity and also impacts the interpretation of studies employing food restriction in behavioral analyses and learning paradigms.

Strengths:

The manuscript is, in general, very clear and the conclusions are straightforward. The experiments are performed in the same conditions for males and females and the authors did not find differences in the behavioral states of male and female mice that could explain differences in energy consumption. Moreover, they show that visual cortex in both males and females does not change its baseline energy consumption in the dark, therefore the adjustment of energy budget in males only targets visual processing.

Weaknesses:

The number of experiments is insufficient to compare the effects of food restriction in males and females directly, which is discussed by the authors: to address this point they use Bayes factor analysis to provide an estimate of the likelihood that females and males indeed differ in terms of energy metabolism and sensory processing adaptions during food restriction.

---

## [Referee Report · Reviewer #2 (Public Review)]

Summary:

Padamsey et al build up on previous significant work from the same group which demonstrated robust changes in the visual cortex in male mice from long-term (2-3 weeks) food restriction. Here, the authors extend this finding and reveal striking sex-specific differences in the way the brain responds to food restriction. The measures included the whole-body measure of serum leptin levels, and V1-specific measures of activity of key molecular players (AMPK and PPARα), gene expression patterns, ATP usage in V1, and the sharpness of visual stimulus encoding (orientation tuning). All measures supported the conclusion that the female mouse brain (unlike in males) does not change its energy usage and cortical functional properties on comparable food restriction.

While the effect of food restriction on more peripheral tissue such as muscle and bones has been well studied, this result contributes to our understanding of how the brain responds to food restriction. This result is particularly significant given that the brain consumes a large fraction of the body's energy consumption (20%), with the cortex accounting for half of that amount. The sex-specific differences found here are also relevant for studies using food restriction to investigate cortical function.

Strengths:

The study uses a wide range of approaches mentioned above which converge on the same conclusion, strengthening the core claim of the study.

Weaknesses:

Since the absence of a significant effect does not prove the absence of any changes, the study cannot claim that the female mouse brain does not change in response to food restriction. However, the authors do not make this claim. Instead, they make the well-supported claim that there is a sex-specific difference in the response of V1 to food restriction.

---

## [Referee Report · Reviewer #3 (Public Review)]

Summary:

The authors food-deprived male and female mice and observed a much stronger reduction of leptin levels, energy consumption in the visual cortex, and visual coding performance in males than females. This indicates a sex-specific strategy for the regulation of the energy budget in the face of low food availability.

Strengths:

This study extends a previous study demonstrating the effect of food deprivation on visual processing in males, by providing a set of clear experimental results, demonstrating the sex-specific difference. It also provides hypotheses about the strategy used by females to reduce energy budget based on the literature.

Weaknesses:

The authors do not provide evidence that females are not impacted by visually guided behaviors contrary to what was shown in males in the previous study.

---

## [Author Response]

The following is the authors’ response to the original reviews.

**Reviewer 1:**
(1) For a number of experiments the authors use their new data set on females and compare that with the data set previously published on males. In how far are these data sets comparable? Have they been performed originally in parallel for example using siblings of different sexes or have the experiments been conducted several years apart from each other? What is the expected variability, if one repeated these experiments with the same sex considering the differences/similarities between experimental setups, housing conditions, interindividual differences, etc.?

This is an important point. We did our best to collect the data in similar conditions (same set-ups; same animal housing conditions) and in experimental cohorts including both males and females. While some data from males were published first, the acquisition of male and female data was done in the same time period.

Specifically, all results shown in Figure 1 and Figure 2 (Serum leptin, PPARalpha, AMPK, RNAseq) come from samples (from both males and females) that were processed at the same time and in similar conditions, by the same authors (Z.P. and P. M.).

For the in vivo data (Figure 3, Supplementary figure 1), the male and female data were collected within a 1–2-year timeframe, in the same setups, by the same two authors (Z.P., D.K.). The males and females were housed under similar conditions (same room, same cage type, in groups of 25). We did not use siblings of different sexes. Independent cohorts (1-12 months apart), including both males and females, went into each data set. The within cohort variability does not obviously differ from between cohort variability, however the n number of animals is too small to confirm this with sufficient statistical power.

Altogether, the differences observed between male and female data cannot be explained by the timing and conditions of data acquisition from both sexes.

(2) Energy consumption and visual processing may differ between periods in which animals are in different behavioral states. Is there a possibility that male and female mice differed in behavioral state during measurements? Were animals running or resting during visual stimulation and during ATP measurements?

We thank the reviewer for this suggestion. We have now edited the text and included a new supplementary figure. All in vivo experiments were done in stationary animals that were resting in a cardboard tube both during 2-photon imaging and ATP measurements. Animals were also well habituated to the setup. In addition, we have imaged pupil diameters during in vivo imaging session. We have quantified pupil diameter during visual stimulation and do not find a sex difference (Supplemental Figure 2). Thus, we did not find a significant difference in behavioural or attentional state between sexes, in our experimental conditions.

We have edited the text to include this information (lines 183-185).

(3) Related to the previous point: the authors show that ATP consumption was reduced in male mice during visual stimulation. What about visual cortex ATP consumption in the absence of visual stimulation? Do food-deprived males and/or females show lower ATP consumption in the visual cortex e.g. during sleep?

We have repeated V1 ATP imaging experiments in the dark, in the absence of visual stimulation, in both males and females (Supplementary figure 1). ATP consumption rates are slower in the dark *vs.* during visual stimulation. Moreover, we find that in the dark, there is no difference in ATP consumption rate between control and food restricted animals of either sex. Thus, the reduced ATP consumption we found with food restriction in males is related specifically to the active processing of visual information.

We have edited the text to include this information (lines 158-159).

**Reviewer 2:**
(1) It appears that the authors have the data for doing decoding analysis, similar to Fig 6D in their previous paper. However, this analysis has not been done for this study. This would be good to include. If the authors have attempted the behavioural discrimination tests on female mice as in the previous study, this would also be useful to include.

The first point of the reviewer is about datasets acquired in males that are included in our previous publication (Padamsey et al., 2022) but not compared to female data in the present manuscript.

Whilst we fully agree that these results would be very useful, we did not have the resources (in terms of skilled researcher and funding) to perform these experiments in female mice. That is why these results are not included in this manuscript.

(2) There appears to be an inconsistency in the methods of reporting OSI. It states that the OSI of grating-responsive neurons was calculated as 1 - circular variance. But then OSI is defined as simply abs(). Also, it would be good to be consistent about reporting medians as the median without confounding with the average (which is the mean). Sentences such as the following do not make sense: The average OSI for an animal was taken as the median OSI value calculated across neurons. This should be corrected throughout the manuscript, where the average is mentioned but the median is measured.

We thank the reviewer for noting this issue and we apologize for the confusion. We have now clarified the above in the manuscript (lines 587-603) and insert the following reference for the detailed explanation of OSI and DSI calculation: Mazurek M, Kager M, Van Hooser SD. Robust quantification of orientation selectivity and direction selectivity. Front Neural Circuits. 2014. https://doi.org/10.3389/fncir.2014.00092

In the figure showing the orientation tuning, the authors have collapsed the two directions of each orientation together. However, if I understand correctly, the calculation of OSI does not do this step of collapsing. In this case, and in the interest of revealing more useful features of the data instead of averaging them out, it would be good to show the average tuning curves with and without FR for all directions, not collapsed.

As with orientation tuning, we found that direction tuning is reduced with food restriction, and that this is significant in males, but not in females. These results are now included in the text, with statistics (lines 179-180) and in Supplemental Figure 3.

**Reviewer 3:**
l. 183-187 The discussion based on the idea that "The Bayes factor analysis helps to differentiate the absence of evidence from the evidence of absence." does not seem very helpful. Using a statistical criterium makes less sense than providing the reader with an estimate largest effect size (if there is any) that is compatible with the observation. If there would be a significant effect but of a very small size would it change the authors' conclusion? That seems unlikely. I recommend removing the sentence on line 184, which is in fact not used afterwards.

We agree with the reviewer. We have now removed the sentence and rephrased (lines 202-208).

**Editor's note:**
Should you choose to revise your manuscript, please include full statistical reporting including exact pvalues wherever possible alongside the summary statistics (test statistic and df) and 95% confidence intervals. These should be reported for all key questions and not only when the p-value is less than 0.05.

We now provide exact p-values alongside the summary statistics (test statistic and df) and 95% confidence intervals for all key results.